# The Effect of Cholesterol Efflux on Endothelial Dysfunction Caused by Oxidative Stress

**DOI:** 10.3390/ijms24065939

**Published:** 2023-03-21

**Authors:** Hua Ye, Qian Liu, Yuanyuan Wang, Ximian Zhen, Nianlong Yan

**Affiliations:** Department of Biochemistry and Molecular Biology, School of Basic Medical Science, Nanchang University, Nanchang 330006, China; huaye@email.ncu.edu.cn (H.Y.); lliucandice@yeah.net (Q.L.); yuanyuanwang@email.nuc.edu.cn (Y.W.); ximianzheng@email.ncu.edu.cn (X.Z.)

**Keywords:** atherosclerosis, endothelial dysfunction, cholesterol efflux, Wnt/β-catenin pathway, endoplasmic reticulum stress

## Abstract

Endothelial dysfunction (ED) is the initiation of atherosclerosis (AS). Our previous studies have found that cholesterol metabolism and the Wnt/β-catenin pathway can affect endoplasmic reticulum stress (ER stress), which ultimately leads to ED. However, the effects of cholesterol efflux on ED, which are caused by oxidative stress and the correlation among ER stress, Wnt/β-catenin pathway, and cholesterol efflux, are not clear during ED. To uncover them, the expressions of liver X receptors (LXRα and LXRβ) and ATP-binding cassette protein A1 (ABCA1) and G1 (ABCG1) in HUVECs (human umbilical vein endothelial cells) were measured under oxidative stress. Moreover, HUVECs were treated with LXR-623 (LXR agonist), cholesterol, tunicamycin, and salinomycin alone or together. The results indicated that oxidative stress-induced ED could deregulate the expressions of LXRα and LXRβ and trigger the ER stress and Wnt/β-catenin pathway, resulting thereafter in the accumulation of cholesterol. Furthermore, similar results were shown after treatment with cholesterol; however, the activation of liver X receptor (LXR) could reverse these changes. Furthermore, other results demonstrated that tunicamycin-induced ER stress could stimulate the accumulation of cholesterol and the Wnt/β-catenin pathway, further leading to ED. Inversely, salinomycin could reverse the above effects by deregulating the Wnt/β-catenin pathway. Collectively, our results showed that cholesterol efflux is partly responsible for the oxidative stress-induced ED; in addition, ER stress, the Wnt/β-catenin pathway, and cholesterol metabolism can interact with each other to promote ED.

## 1. Introduction

Atherosclerosis (AS), one of the main causes of mortality globally, is the primary factor of cardiovascular diseases and contributes to the abnormal functions of the heart, brain, and kidney and to peripheral vascular lesions [1,2,3,4]. Numerous studies have shown that endothelial dysfunction (ED) is a marker during the initial stage of AS, which can cause poor prognoses of vascular occlusion, plaque shedding, and rupture [5,6]. Moreover, ED is a dysfunctional phenomenon of endothelial cells under adverse stimulations such as oxidative stress [7,8,9]. Therefore, since oxidative stress is crucial to ED [10], in this study, the activated ED model of human umbilical vein endothelial cells (HUVECs) was established by treatment with H_2_O_2_ [11].

The endoplasmic reticulum (ER) is an organelle with functions such as protein folding, calcium ion storage, and the synthesis of lipids and carbohydrates [12], which also regulates cellular metabolism. Therefore, protein overload, calcium ion imbalance, oxidative stress, and other factors may cause physiological dysfunctions of the ER, which further induce unfolded protein responses (UPRs) to maintain ER homeostasis [13,14,15]. If the homeostasis cannot be restored, ER stress will be triggered [13,14,15]. Moreover, the vital role of the ER on ED has been confirmed by many studies. For instance, Matshediso Zachariah et al. found that the high concentrations of selenium in endothelial cells can trigger ER stress, which leads to ED, through the increasing of ROS [10]. Melatonin inhibited JNK/Mff signaling and ER stress and hence protected endothelial cells against ox-LDL-induced damage [16]. In addition, abnormal lipid metabolism has been verified to be significant to facilitate ER stress. Lipotoxicity can damage liver cells by inducing ER stress in non-alcoholic fatty liver [17,18,19,20], and high-fat diets in mice or ectopic lipid deposition of cells can activate ER stress in the liver and inhibit insulin receptor signaling to accelerate the development of steatohepatitis [21,22]. Furthermore, damaged cholesterol metabolism also can trigger ER stress and ED in the endothelial cells [23]. It has been shown that elevated total cholesterol and cholesterol crystals can increase arginases metabolism and impair nitric oxide signaling, further aggravating ED in gestational diabetes mellitus and early atherosclerosis [24,25]. Moreover, our previous study indicated that simvastatin, an inhibitor of HMGCR (3-hydroxy-3-methylglutaryl coenzyme A reductase) in the biosynthesis of cholesterol, could alleviate oxidative stress-induced ED by inhibiting ER stress [26]. Moreover, SMS2 (sphingomyelin synthase 2) has been demonstrated to promote the intracellular accumulation of cholesterol, which results in ER stress and oxidative stress-induced ED [23]. To sum up, both ER stress and cholesterol metabolism are involved in the occurrence of ED.

The Wnt/β-catenin pathway is another inducement to ED. For instance, increased serum leptin levels in chronic kidney patients can lead to ED via the MTA1-WNT/β-catenin pathway [27]. Under oxidative stress, inhibition of the Wnt/β-catenin pathway has been shown to mitigate ED [28]. Moreover, it has been discovered that simvastatin can suppress the Wnt/β-catenin pathway by downregulating intracellular cholesterol accumulation, further resulting in relieving ED [26]. Moreover, our previous research also found that SMS2 could augment oxidative stress-induced ED by enhancing the Wnt/β-catenin pathway [23]. Obviously, the Wnt/β-catenin pathway, ER stress, and cholesterol metabolism all participate in ED. However, the relationship among the above three and the effects of cholesterol efflux on oxidative stress-induced ED need to be further investigated. Thus, this research aimed to explore the role and possible mechanisms of cholesterol efflux on oxidative stress-induced ED in HUVECs.

## 2. Results

### 2.1. Oxidative Stress Inhibited the Cholesterol Efflux of HUVECs

Since both cholesterol metabolism and oxidative stress are associated with ED, HUVECs were treated with H_2_O_2_ to explore whether the oxidative stress could affect the cholesterol metabolism of HUVECs. The results (Figure 1A) showed that the expression of cholesterol efflux-related proteins ABCA1 and their transcription factor LXRα and LXRβ were significantly downregulated in the treatment group. Moreover, HMGCR participated in cholesterol synthesis and was significantly upregulated under treatment (Figure 1A,B). Furthermore, the result (Figure 1C) of filipin staining demonstrated that the level of intracellular cholesterol was increased with H_2_O_2_ treatment. These outcomes indicated that oxidative stress could facilitate the accumulation of intracellular cholesterol by reducing the efflux and increasing the synthesis of cholesterol. However, our previous studies showed that simvastatin could attenuate the oxidative stress-induced ED of HUVECs [26]; therefore, the influence of cholesterol efflux on ED was to be investigated later.

### 2.2. The Cholesterol Metabolism Affected the Cholesterol Accumulation in HUVECs 

To further explore the effects of cholesterol efflux on endothelial cells, the activated LXR model of HUVECs was established by treatment with LXR-623 (activated both LXRα and LXRβ). The results indicated that LXR was activated successfully, and the impacts of H_2_O_2_ could be inhibited by upregulating ABCA1 and ABCG1 in the treatment group (Figure 2A). Additionally, filipin staining suggested that the H_2_O_2_-induced accumulation of intracellular cholesterol could be reversed by the treatment of LXR-623 (Figure 2B). These results proved that the enhanced cholesterol efflux can alleviate the accumulation of cholesterol in HUVECs caused by oxidative stress. Moreover, cholesterol was directly added to detect the effects of cholesterol metabolism on the accumulation of cholesterol. Interestingly, the results (Figure 2C) found that the expression of ABCG1 was significantly downregulated, and the level of ABCA1 was upregulated. Moreover, the content of total intracellular cholesterol was elevated after treatment with cholesterol alone (Figure 2C,D). However, when under the treatment of both H_2_O_2_ and cholesterol, the variations of ABCA1, ABCG1, and total intracellular cholesterol were not significantly different from the results after being treated separately in cholesterol or H_2_O_2_ (Figure 2C,D). The outcomes showed that the cholesterol treatment alone can also induce the accumulation of cholesterol in HUVECs. However, there were no synergistic effects in the H_2_O_2_ + CHOL group.

### 2.3. Cholesterol Accumulation Can Enhance the Damage of HUVECs Induced by Oxidative Stress

To explore endothelial cell damage after altering the cholesterol metabolism, the levels of LDH, NOS, SOD, and ROS in HUVECs were measured. The studies discovered that the contents of LDH and ROS were upregulated by the treatment of H_2_O_2_, and the extracellular LDH and intracellular ROS were decreased in the LXR-623-treated group (Figure 3A,D). Conversely, the levels of intracellular SOD and NOS showed the opposite trend (Figure 3B,C). These findings indicated that oxidative stress-induced HUVEC injury and cholesterol efflux can alleviate HUVECs cell damage caused by oxidative stress. Furthermore, the levels of extracellular LDH and intracellular ROS were upregulated, and intracellular SOD and NOS were downregulated in the cholesterol treated group, which was similar with the results of H_2_O_2_ treatment. However, there were no significant variations in LDH, ROS, SOD, and NOS under both treatments of H_2_O_2_ and cholesterol compared to H_2_O_2_ or cholesterol treatment alone (Figure 3E–H). To sum up, the accumulation of cholesterol by treating cholesterol directly can enhance the injury of HUVECs.

### 2.4. Cholesterol Accumulation under Oxidative Stress can Increase the Adhesion Ability of HUVECs

The role of cholesterol accumulation on the adhesion capacity of HUVECs was investigated. The results of Western blotting showed that the levels of adhesion molecules ICAM-1(Intercellular cell adhesion molecular-1), VCAM-1(Vascular cell adhesion molecular-1), and MCP-1(Monocyte chemoattractant protein-1) were upregulated under H_2_O_2_-induced oxidative stress, which enhanced the adhesion capacity of HUVECs to THP-1 (Human myeloid leukemia mononuclear cells). Nevertheless, there were opposite results in the LXR-623 treated group (Figure 4A,B). Therefore, it was suggested that the H_2_O_2_-induced adhesion capacity of HUVECs was attenuated by the cholesterol effluxion. Consistent with the above results, the expressions of ICAM-1, VCAM-1, and MCP-1 were also upregulated through cholesterol treatment, resulting in the elevation of the adhesion capacity of HUVECs. However, with the treatment of both H_2_O_2_ and cholesterol, the variations of ICAM-1, VCAM-1, and MCP-1 were slightly downregulated compared with those detected with the H_2_O_2_ or cholesterol treatment alone, and similar results were also observed in the adhesion capacity of HUVECs to THP-1 (Figure 4C,D). In general, these data could show that the adhesion ability of HUVECs could be promoted by treating the cholesterol alone.

### 2.5. The Accumulation of Cholesterol in HUVECs Triggers ER Stress and Activates the Wnt/β-Catenin Pathway

As our previous studies indicated, the Wnt/β-catenin pathway is closely related with ED [29], and ER stress can cause ED via activation of the Wnt/β-catenin pathway [23]. Therefore, the ER stress markers Glucose-regulated protein 78 (GRP78) and C/EBP homologous protein (CHOP), β-catenin, and p-β-catenin were detected. The results determined that the expressions of GRP78, CHOP, and β-catenin were upregulated, and p-β-catenin was downregulated in the treatment of H_2_O_2_, whereas LXR-623 reversed the above trends with the decreased expressions of GRP78, CHOP, and β-catenin and the increased level of p-β-catenin protein (Figure 5A,B). These findings demonstrated that the enhanced cholesterol efflux can suppress ER stress and the Wnt/β-catenin pathway caused by the accumulation of cholesterol under oxidative stress. Moreover, the expressions of GRP78, CHOP, and β-catenin were enhanced, and the expression of p-β-catenin was inhibited under the treatment of cholesterol alone. However, the results of treatment with H_2_O_2_ and cholesterol together were not significantly different compared with the treatment of H_2_O_2_ or cholesterol alone (Figure 5C,D). In general, it has been shown that both the ER stress and Wnt/β-catenin pathway can be triggered by treating cholesterol directly.

### 2.6. Wnt/β-Catenin Pathway Can Interact with ER Stress, and Both of Them Can Affect the Cholesterol Metabolism

It has been confirmed that ER stress, the Wnt/β-catenin pathway, and cholesterol accumulation play critical roles in ED, and cholesterol accumulation can trigger ER stress and the Wnt/β-catenin pathway. Therefore, whether ER stress and the Wnt/β-catenin pathway can modulate the cholesterol accumulation was explored. HUVECs were treated with tunicamycin to stimulate ER stress, and salinomycin was used to inhibit the Wnt/β-catenin pathway. The results (Figure 6A) verified that the expressions of GRP78 and CHOP were upregulated under tunicamycin treatment and downregulated by salinomycin, respectively. In other words, the effects of tunicamycin on HUVECs were reversed by salinomycin (Figure 6A). The findings demonstrated that the inhibition of the Wnt/β-catenin pathway can suppress ER stress. Furthermore, the influences of ER stress on the Wnt/β-catenin pathway were determined later. The data indicated that tunicamycin could enhance the salinomycin-induced downregulation of β-catenin expression; moreover, p-β-catenin presented the opposite trend (Figure 6B). Hence, it can be suggested that enhanced ER stress can activate the Wnt/β-catenin pathway. Moreover, the influences of ER stress and the Wnt/β-catenin pathway on the cholesterol efflux revealed that tunicamycin-induced ER stress could downregulate the expressions of ABCA1 and ABCG1, which resulted in an increase in total intracellular cholesterol. Conversely, salinomycin, as an inhibitor of the Wnt/β-catenin pathway, could change over the tunicamycin-induced cholesterol accumulation by upregulating ABCA1 and ABCG1 expressions (Figure 6C,D). In summary, both ER stress and the Wnt/β-catenin pathway could promote intracellular cholesterol accumulation.

### 2.7. Inhibiting the Wnt/β-Catenin Pathway under Endoplasmic Reticulum Stress Can Decrease HUVECs Cell Damage and the Adhesion Capacity

To further investigate the functions of ER stress and the Wnt/β-catenin pathway to ED, the relevant indicators of ED were determined through treatment with tunicamycin to stimulate ER stress and salinomycin to inhibit the Wnt/β-catenin pathway. The results suggested that extracellular LDH and intracellular ROS were elevated under the treatment of tunicamycin alone, which were reduced by salinomycin treatment (Figure 7A,D). Conversely, the expressions of intracellular SOD and NOS had the opposite trend (Figure 7B,C). These results indicated that the activation of ER stress and the Wnt/β-catenin pathway can induce HUVEC injury. Furthermore, the expressions of ICAM-1, VCAM-1, and MCP-1 and the adhesion capacity of HUVECs to THP-1 were upregulated under tunicamycin treatment, which were reversed under salinomycin treatment (Figure 7E,F). These outcomes proved that the ER stress and Wnt/β-catenin pathway can augment the HUVECs adhesion capacity.

Salinomycin, a potent inhibitor of Wnt/β-catenin signaling, is able to block the phosphorylation of Wnt-induced LRP6 (Recombinant Low Density Lipoprotein Receptor Related Protein 6). Next, LRP6 was exclusively knocked down using shRNA interference to further explore the effects of Wnt/β-catenin signaling on the ER stress and cholesterol metabolism (Figure 8A). Interestingly, specific knockdown of LRP6 did obstruct Wnt/β-catenin signaling, ultimately resulting in decreased ER stress and cholesterol accumulation (Figure 8B–E). All in all, knocking down LRP6 had the same effects as salinomycin, that is, mitigating the effects of tunicamycin on endothelial cells, finally alleviating ER stress, and promoting cholesterol outflow.

## 3. Discussion

In this research, we discovered that the optimum concentration of H_2_O_2_ was 500 μmol/L. However, Ransy et al. believed that this dose could release O_2_ from the catalase reaction, which was disproportionally high with regard to physiological oxygen concentration [30]. A high concentration of oxygen may cause other reactions in cells. However, in our studies, the damage levels of HUVECs were raised with the increase of H_2_O_2_ concentration (from 100 to 700μmol/L) for the secretion of LDH. Therefore, the mechanisms of HUVEC injury are very complicated, which may be related to the Fenton’s reaction, a large excess in antioxidant defense, etc. [30]. More importantly, in this concentration, we found that the expressions of protein related to cholesterol metabolism changed, caused by H_2_O_2_. Among them, the levels of cholesterol efflux-related proteins (ABCA1 and ABCG1) were significantly downregulated, and cholesterol synthesis-related protein HMGCR was upregulated (Figure 1). Furthermore, the expressions of LXRα and LXRβ were decreased, which are the transcript factors of ABCA1 and ABCG1. Obviously, all of these variations contributed to the accumulation of intracellular cholesterol (Figure 1) and partly indicated that the cholesterol efflux may participate in the oxidative stress-induced ED. In order to further verify the functions of cholesterol efflux in the oxidative stress induced-ED, the activity of LXR was elevated by LXR-623 under oxidative stress. Results showed that both the accumulation of cholesterol and ED were decreased (Figure 2, Figure 3 and Figure 4). These demonstrated that the enhanced cholesterol efflux with LXR-623 treatment can be attenuated by oxidative stress induced-ED, which directly reduced the accumulation of cholesterol by LXR. In other words, the accumulation of cholesterol may be one of the direct factors to induce ED (Figure 2, Figure 3 and Figure 4). In patients with coronary ED, Monette et al. found similar results to ours, which showed that the cholesterol efflux capacity was a strong, inversed predictor of ED [31]. Subsequently, to prove the above hypothesis, the content of cholesterol was increased by adding cholesterol directly in this study. These results indicated that enhancing the level of cholesterol caused the accumulation of cholesterol and ED, which was similar to the observations of the oxidative stress-induced ED; therefore, it was suggested that cholesterol accumulation is a possible main factor of ED (Figure 2, Figure 3 and Figure 4). Interestingly, when HUVECs were treated with cholesterol alone, the expressions of ABCA1 and ABCG1 had the opposite results and finally affected the accumulation of cholesterol (Figure 2C,D). These results showed the different functions of ABCA1 and ABCG1 in endothelial cells. Moreover, the treatment of both H_2_O_2_ and cholesterol did not acquire a synergistic effector superposition effect on enhancing ED and the adhesion capacity of HUVECs to THP-1. Contrarily, when the H_2_O_2_ + CHOL group was compared to the H_2_O_2_ or CHOL group alone, ED (LDH and NOS) and the expressions of ICAM-1, VCAM-1, and MCP-1 had no significant difference and showed slight downregulation. The reasons may be related to the feedback regulation, because cholesterol was treated earlier than H_2_O_2_ by about 2 h (Figure 4C,D). Additionally, our previous experiment revealed that the inhibition of HMGCR by simvastatin could suppress the synthesis and accumulation of intracellular cholesterol and ultimately alleviated ED in HUVECs, which affirmed that cholesterol metabolism is an important factor for ED once again [26]. Moreover, we also found that SR-B1 and LDLR were decreased under oxidative stress-induced ED, which might be regulated by the cholesterol accumulation and also illustrated that cholesterol metabolism is associated with oxidative stress-induced ED. Clearly, all of these studies indicated that cholesterol metabolism plays a crucial role in the process of ED.

The influence of cholesterol metabolism on ED is possibly related to ER stress, which promotes ED [23,26]. The ER is involved in the biosynthesis and storage of cholesterol; therefore, the cholesterol metabolism also can stimulate ER stress. Jiansen Yan et al. discovered that ER stress could be activated with cholesterol treatment by stimulating mSREBP1 in IDD, inducing the pyroptosis in NP cells and ECM degradation [32]. Moreover, our previous study indicated that the activating LXRα, which promoted cholesterol efflux by augmenting ABCA1 and ABCG1 expressions, could significantly reverse tunicamycin-induced ER stress in hepatocytes and macrophages [20]. Moreover, cholesterol homeostasis was also revealed to modulate the epithelial–mesenchymal transition (EMT) by regulating the ER stress in breast cancer cells [33]. Furthermore, in the occurrence of ED and cardiovascular disturbances, ER stress is a critical mechanism of ED [23,26,34]. In HUVECs, we previously confirmed that simvastatin attenuated the accumulation of cholesterol, which mitigated ED by relieving ER stress. Conversely, SMS2 could promote the accumulation of intracellular cholesterol, which leads to ER stress and ED [23,26]. In this study, when the HUVECs were treated with cholesterol and LXR-623, the ER stress was positively correlated with cholesterol accumulation and ED (Figure 5). Although the H_2_O_2_ + CHOL group still did not acquire a synergistic effect or superposition effect on the ER stress, which showed the same cause as the above, these results suggested that ED affected by the cholesterol efflux may depend on ER stress (Figure 5). 

The Wnt/β-catenin pathway is one of the factors that can promote ED. In this study, we discovered that changing cholesterol metabolism could regulate the Wnt/β-catenin pathway, which was also positively correlated with cholesterol accumulation and ED (Figure 5). Mechanistically, firstly, the regulation of cholesterol metabolism to the Wnt/β-catenin signaling pathway depended on the LRP6, which is a coreceptor of Wnt and located on the lipid rafts [35,36]. Lipid rafts are functional areas on the cell membrane, the major components of which are sphingolipids and cholesterol [36]. Thus, cholesterol efflux can impact the transmembrane signaling of LRP6 and further affect the Wnt/β-catenin pathway via changing the cholesterol compositions in the lipid raft. Secondly, ER stress was proved to be involved in the regulation of the Wnt/β-catenin pathway. For example, research showed that ER stress can inhibit the Wnt/β-catenin pathway in cancers cells [37,38,39]. Therefore, it was verified that ER stress can block the Wnt/β-catenin pathway, causing cancer cell apoptosis [37,38]. Contrarily, when HUVECs were treated with tunicamycin to trigger ER stress, the results showed that ER stress can activate the Wnt/β-catenin pathway and promote cholesterol accumulation by downregulating ABCA1 and ABCG1 expressions (Figure 6). Apparently, the results (1) revealed that the cholesterol accumulation induced by ER stress was partly alleviated by cholesterol efflux, and (2) were contradictory to the studies in cancers cells, possibly due to cell types. As a result, the Wnt/β-catenin pathway has different effects on endothelial cells and cancer cells. Moreover, our results also indicated that ER stress could regulate cholesterol metabolism (Figure 6). In male zebrafish, Zhang et al. showed that cholesterol genes were involved in ER stress and upregulated cholesterol gene expressions to bring about hepatic lipid accumulation by MC-LR [40]. Mechanistically, sterol regulatory element-binding protein (SREBP), a major component related to the cholesterol metabolism, can be activated and regulated by ER stress [41]. Obviously, both our present and previous studies suggested that cholesterol metabolism can activate ER stress and the Wnt/β-catenin pathway, further accelerating the process of ED (Figure 5) [23,26]. On the contrary, ER stress can regulate the cholesterol metabolism and Wnt/β-catenin pathway as well. However, what needs further discovery is whether the Wnt/β-catenin pathway can modulate ER stress and cholesterol metabolism. Therefore, when we inhibited the Wnt/β-catenin pathway treated with salinomycin or knocked down the expression of LRP6, the outcomes revealed that both the ER stress and cholesterol accumulation were decreased, and the regulation of cholesterol was related to ABCA1 and ABCG1 (Figure 6). Interestingly, the combined treatment with tunicamycin and salinomycin (or knock down of the expression of LRP6) can antagonize the activity of each other to HUVECs (Figure 6, Figure 7 and Figure 8). It was obvious that (1) the Wnt/β-catenin pathway simultaneously can regulate ER stress and cholesterol efflux, and (2) the Wnt/β-catenin pathway and ER stress can affect each other. 

## 4. Materials and Methods

### 4.1. Cell Culture

HUVECs (Cell Bank of Type Culture Collection of the Chinese Academy of Sciences, Shanghai, China), within 30 generations, were cultured with DMEM medium (11995, Solaibio, Beijing, China), and the THP-1 cells (Cell Bank of Type Culture Collection of the Chinese Academy of Sciences, Shanghai, China) were incubated with RPMI-1640 medium (31800, Solaibio, Beijing, China). All media were filled with 10% FBS, and all cell cultures were carried out in an environment at 37 °C containing 5% CO_2_, H_2_O_2_ (500 μmol/L, Guangdong Hengjian Pharmaceutical Co., Ltd., Jiangmen, China), LXR-623 (5 μmol/L, HY10629, MedchemExpress, Shanghai, China), cholesterol (100 μmol/L, C8280, Solaibio, Beijing, China), tunicamycin (10 μmol/L, MB5419, Meilunbio, Dalian, China), and salinomycin (5 μmol/L, HY-15597, MedchemExpress, Shanghai, China).

### 4.2. Cell Transfection

The LRP6 gene was cloned into the pLKO.1 puro vector, and further the resulting vector was transferred into TransStbl3 Chemically Competent Cells (CD521-01, TransGen Biotech, Beijing, China). The plasmid was extracted with an EasyPure HiPure Plasmid MiniPrep Kit (EM111-01, TransGen Biotech, China) according to the instructions. We cotransfected psPAX.2, pMD2.G and the target plasmid into 293 T cells. TurboFect (R0531, Thermo Scientific, China) was used to transfect cells according to the protocol. The lentivirus in the supernatant of the medium was concentrated after culturing for 48 h using Lentivirus Concentration Kit (FV101-01, TransGen, Beijing, China). Finally, the concentrated lentivirus was used to infect HUVECs at the confluency of 50% for 48 h. Two pairs of shRNA were created. One of the LRP6 shRNA sequences was 5′-CCGATGCAATGGAGATGCAAA-3′, and the other was 5′-GATAGCCTTCAGTTAACTAAC-3′.

### 4.3. Filipin Staining

HUVECs were seeded on 24-well plates and cultured into an incubator at a density of 2 × 10^3^ per well. With cell confluency reaching about 60%, HUVECs were treated with drugs and cultured for 24 h. Afterwards, we discarded the medium and then washed the cells with PBS buffer. Then, after being fixed with 4% paraformaldehyde at 37 °C for 35 min, HUVECs were rinsed three times with PBS buffer and further immersed in glycine (1.5 mg/mL) for 10 min to neutralize the remaining formaldehyde. Finally, after rinsing off the excess dye with PBS after dyeing with filipin solution (50 μg/mL, F31601, Harveybio, Beijing, China) for 2 h, we observed the fluorescence intensity using a LeicaDM1000 (magnification, 5×) and an Olympus FV3000 (magnification, 10×) and then calculated the average fluorescence intensity of all the signal regions of the entire image by Image J 1.53e software, excluding the black background area.

### 4.4. Assessment of ED by Measuring the Content of LDH, SOD, and NOS

HUVECs were planted in 6-well plates and then cultured with drugs together for 24 h with the cell density attaining about 60%. The supernatant was taken for the determination of LDH activity by a lactate dehydrogenase (LDH) assay kit (A020-1-2, Nanjing Jiancheng Bioengineering Institute, Nanjing, China). Furthermore, HUVECs homogenated with PBS buffer were taken for the determination of SOD and NOS activity. The superoxide dismutase (SOD) assay kit (A001-3-2, Nanjing Jiancheng Bioengineering Institute, Nanjing, China) was used to test SOD content, and the nitric oxide synthase (NOS) assay kit (A014-2-2, Nanjing Jiancheng Bioengineering Institute, Nanjing, China) was used to detect NOS content. All of the above experiments were measured strictly according to the corresponding instructions.

### 4.5. Measurement of ROS Contents

The level of intracellular ROS was determined by the fluorescence probe 2,7-dichlorodi-hydrofluorescein diacetate (DCFH-DA). HUVECs were tiled in a 24-well plate with a density of 2 × 10^3^ per well. A DCFH-DA working solution (15 μmol/L) was prepared with DCFH-DA stock solution (10 mmol/L, E-BC-K138-F, Elabscience, Wuhan, China) and buffer. With the medium aspirated carefully, the cells were rinsed once with buffer solution. Then, at about 70% growth, the HUVECs and the DCFH-DA working solution were cultured together for 50 min. Finally, when HUVECs were rinsed three times with buffer solution, the fluorescence degree was observed using an Olympus IX71 fluorescence microscope (magnification,10×), and then the mean fluorescence intensities of all the signal regions of the entire picture were analyzed with Image J 1.53e software.

### 4.6. Measurement of Adhesion Ability of HUVECs to THP-1 Cells

The HUVECs were incubated in 24-well plates. Furthermore, 5 × 10^4^ THP-1 cells, labeled with BCECF/AM (B115503, Aladdin, Shanghai, China) for 1 h in the dark cell incubator, were then transferred into drug-treated HUVECs in 24-well plates when the HUVECs had grown to about 70% and cultured for 1 h in the dark. Afterwards, the THP-1 cells and HUVECs were dyed with Hoechst33342 for 40 min after being removed from the previous medium. Finally, PBS buffer was added to rinsed cells three times to gently wash away the THP-1 cells that had not adhered. Images were taken with an Olympus inverted fluorescence microscope (magnification, ×4). The average number of THP-1s of three independent tests was calculated.

### 4.7. Western Blot

The HUVECs were harvested with trypsin (T8150, Solaibio, Beijing, China) after being treated with drugs for 24 h. RIPA lysate (C1053, Applygen Technologies Inc., Beijing, China) was used to lyse cells, and then the total protein of supernatant was collected through centrifugation. The collected supernatant was boiled for 5 min after adding 1 × loading buffer (DL101-02, TransGen Biotech, Beijing, China) and then electrophoretically isolated on 8–12%SDS–PAGE gel. Then we used 5% skimmed milk to block the PVDF membrane for 1 h. Firstly, the PVDF membrane was incubated with the primary antibodies at 4 °C for 12 h and washed with 1 × TBST for 30 min. Next, the PVDF membrane was incubated with the secondary antibodies at 37 °C for 1 h, further washed with 1 × TBST for 30 min, with 10 min each time at 100 rpm/min. At last, ECL ultra-sensitive luminescent liquid was added on the PVDF film equably, which was imaged on the BIORAD gel imager. The antibodies are as follows: MCP-1 (WL02966, Wanlei, Shenyang, China), 1:500; CD106/VCAM-1 (WL02474, Wanlei, Shenyang, China), 1:500; GRP78 (66574-1-Ig, Proteintech, Hubei, China), 1:10000; GAPDH (HRP-60004, Proteintech, Hubei, China), 1:40000; β-catenin (17565-1-AP, Proteintech, Hubei, China), 1:4000; ICAM-1 (10831-1-AP, Proteintech, Hubei, China), 1:1000; ABCA1 (D155299, BBI, Shanghai, China), 1:500; ABCG1 (AP6529A, Abgent, San Diego, USA), 1:500; LXRα (D198471, BBI, Shanghai, China), 1:2000; LXRβ (A04523-2, BOSTER, Hubei, China), 1:1000; LRP6 (AP1191, ABclonal, Hubei, China), 1:1000; HMGCR (AP11955B, Abgent, San Diego, CA, USA), 1:1000 and p-β-catenin (DF2989, Affinity, Jiangsu, China), 1:2000. In these experiments, the protein bands normalized to GAPDH. 

### 4.8. Statistical Analysis

All data were first tested by the Shapiro–Wilk normality test using GraphPad Prism 6.0 software, and we found *p* > 0.1, suggesting that all date conformed to a normal distribution. Then, a Student’s *t*-test was performed to compare between two groups, and one-way analysis of variance (ANOVA) was used to analyze the differences between more than two groups. The fluorescence intensity analysis was processed by Image J 1.53e. All experiments were performed at least three times, with three biological replicates per experiment, and all data were expressed as mean ± standard deviation. *p* < 0.05 indicates there was a statistical difference.

## 5. Conclusions

Our studies suggested that inhibiting the cholesterol efflux of HUVECs could partly induce the cholesterol accumulation by decreasing the activity of LXRs, which led to the oxidative stress-induced ED. Mechanically, both cholesterol accumulation and Wnt/β-catenin pathway in HUVECs can trigger the ER stress; conversely, ER stress can regulate cholesterol metabolism and the Wnt/β-catenin pathway as well. Moreover, cholesterol metabolism and Wnt/β-catenin pathway also can affect each other. In conclusion, ER stress, cholesterol metabolism, and the Wnt/β-catenin pathway can interact with each other to promote ED in HUVECs (Figure 9). 

## Figures and Tables

**Figure 1 ijms-24-05939-f001:**
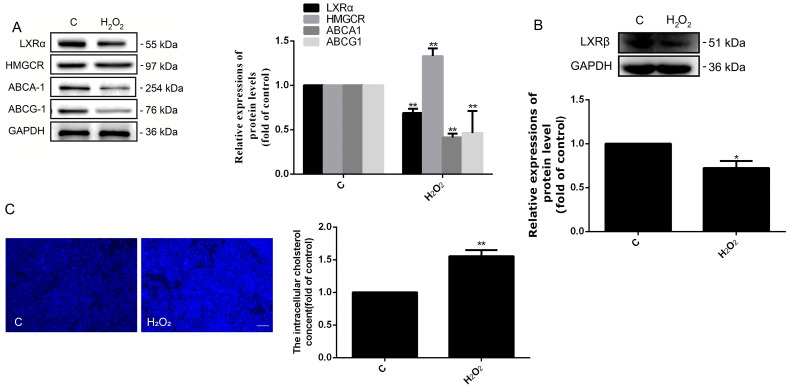
H_2_O_2_ can promote cholesterol accumulation by oxidative stress. (**A**,**B**) The protein levels of LXRα, LXRβ, HMGCR, ABCA1, and ABCG1 were measured by a Western blot analysis. (**C**) The accumulation of ER cholesterol after filipin staining was visualized under a fluorescence microscope (magnification 5×). *n* = 3, * *p* < 0.05, and ** *p* < 0.001 vs. the C group (treated by PBS). C, control group; H_2_O_2_, cells treated with H_2_O_2_ (500 μmol/L) for 24 h. The scale of the pictures was 100 μm.

**Figure 2 ijms-24-05939-f002:**
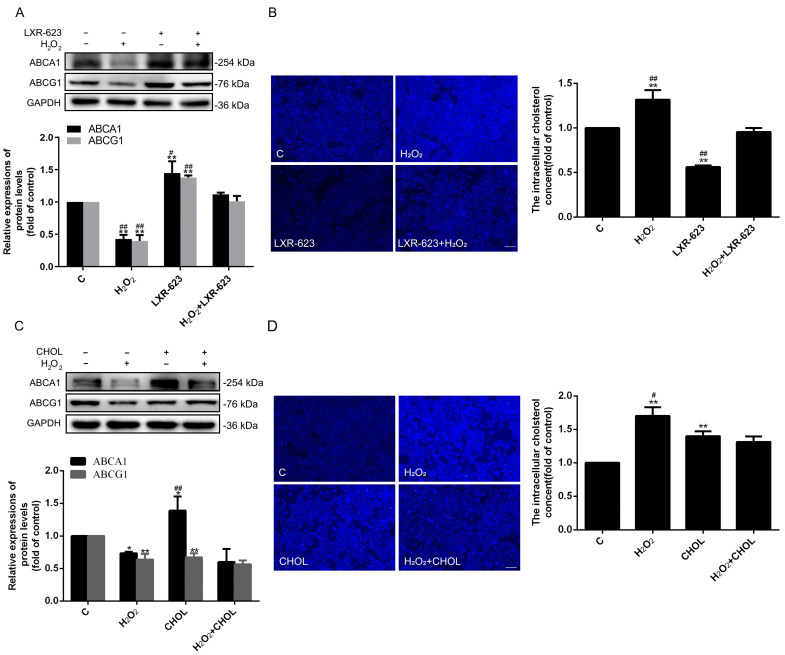
LXR-623 can promote cholesterol efflux, and adding cholesterol extracellularly can induce cholesterol accumulation. (**A**,**C**) The expressions of ABCA1 and ABCG1 were detected by Western blot. (**B**,**D**) Filipin staining detected the intracellular cholesterol of HUVECs (magnification 5×). *n* = 3, * *p* < 0.05, and ** *p* < 0.001 vs. the C group; ^#^ *p* < 0.05, and ^##^ *p* < 0.001 vs. the H_2_O_2_ + LXR-623 group; ^#^ *p* < 0.05, and ^##^ *p* < 0.001 vs. the H_2_O_2_ + CHOL group. The C (control) group of LXR-623 and cholesterol were treated by DMSO and chloroform, respectively; H_2_O_2_, cells treated with H_2_O_2_ (500 μmol/L) for 24 h; LXR-623, cells treated with LXR-623 (5 μmol/L) for 24 h; H_2_O_2_ + LXR-623, cells treated with LXR-623 (5 μmol/L) for 2 h and then treated with H_2_O_2_ (500 µmol/L) for 22 h; CHOL, cells were treated by cholesterol (100 μmol/L) for 24 h; H_2_O_2_ + CHOL, cells were firstly treated by cholesterol (100 μmol/L) for 2 h, and then H_2_O_2_ (500 µmol/L) was used to treat cells for 22 h. The scale of the images is 100 μm.

**Figure 3 ijms-24-05939-f003:**
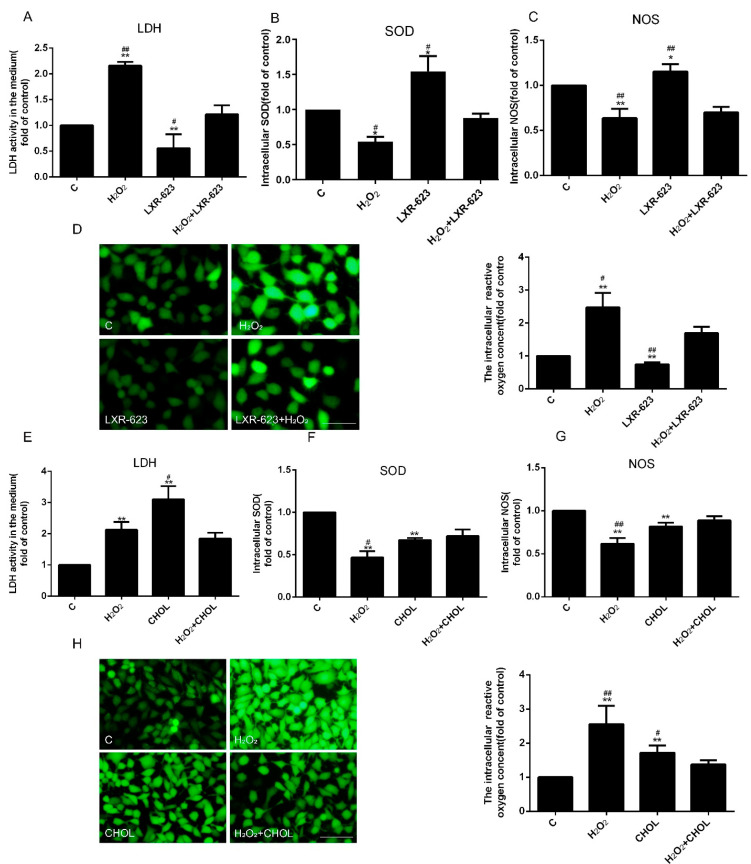
LXR-623 can attenuate H_2_O_2_-induced endothelial cell injury by increasing cholesterol efflux, and adding cholesterol extracellularly can induce cell damage by promoting the deposition of intracellular cholesterol. (**A**) LDH, (**B**) SOD, and (**C**) NOS levels were measured with assay kits. (**D**) The activity of ROS was detected by fluorescemce probe DCFH-DA (15 μmol/L). (**E**) LDH, (**F**) SOD, (**G**) NOS levels were detected with assay kits. (**H**) Using fluorescemce probe DCFH-DA (15 μmol/L) to detect ROS activity (magnification 10×). *n* = 3, * *p* < 0.05, and ** *p* < 0.001 vs. the C group; ^#^ *p* < 0.05 and ^##^ *p* < 0.001 vs. the H_2_O_2_ + LXR-623 group; ^#^ *p* < 0.05 and ^##^ *p* < 0.001 vs. the H_2_O_2_ + CHOL group. The C (control) group of LXR-623 and cholesterol were treated by DMSO and chloroform, respectively; H_2_O_2_, cells treated with H_2_O_2_ (500 μmol/L) for 24 h; LXR-623, cells treated with LXR-623 (5 μmol/L) for 24 h; H_2_O_2_ + LXR-623, cells treated with LXR-623 (5 μmol/L) for 2 h and then treated with H_2_O_2_ (500 µmol/L) for 22 h; CHOL, cells were treated by cholesterol (100 μmol/L) for 24 h; H_2_O_2_ + CHOL, cells were firstly treated by cholesterol (100 μmol/L) for 2 h, and then H_2_O_2_ (500 µmol/L) was used to treat cells for 22 h. The scale of the images is 100 μm.

**Figure 4 ijms-24-05939-f004:**
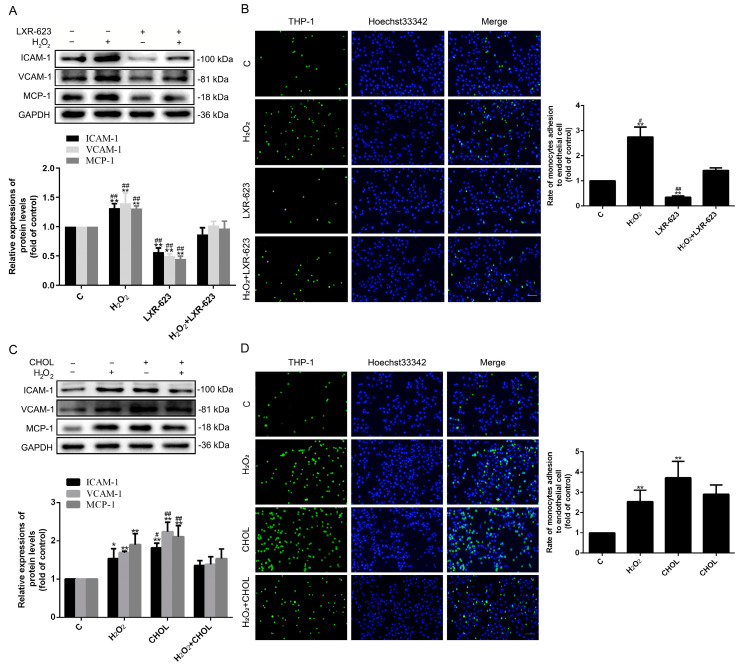
LXR-623 can attenuate the H_2_O_2_-induced adhesion activity of HUVECs by increasing cholesterol efflux, and adding cholesterol extracellularly can augment the adhesion activity of HUVECs by promoting intracellular cholesterol accumulation. (**A**,**C**) The expressions of ICAM-1, VCAM-1, and MCP-1 were detected by Western blot. (**B**,**D**) BCFECF/AM (2',7'-Di-(2-carboxyethyl)-5(6)-carboxy fluorescein) was used to label THP-1. The number of THP-1 attached to HUVEC represents the adhesion capacity of HUVEC. Green fluorescence was labeled THP-1 (magnification, 4×). *n* = 3, * *p* < 0.05 and ** *p* < 0.001 vs. the C group; ^#^ *p* < 0.05 and ^##^ *p* < 0.001 vs. the H_2_O_2_ + LXR-623 group; ^#^ *p* < 0.05 and ^##^ *p* < 0.001 vs. the H_2_O_2_ + CHOL group. The C (control) group of LXR-623 and cholesterol were treated by DMSO and chloroform, respectively; H_2_O_2_, cells treated with H_2_O_2_ (500 μmol/L) for 24 h; LXR-623, cells treated with LXR-623 (5 μmol/L) for 24 h; H_2_O_2_ + LXR-623, cells treated with LXR-623 (5 μmol/L) for 2 h and then treated with H_2_O_2_ (500 µmol/L) for 22 h; CHOL, cells were treated by cholesterol (100 μmol/L) for 24 h; H_2_O_2_ + CHOL, cells were firstly treated by cholesterol (100 μmol/L) for 2 h, and then H_2_O_2_ (500 µmol/L) was used to treat cells for 22 h. The scale of the images is 100 μm.

**Figure 5 ijms-24-05939-f005:**
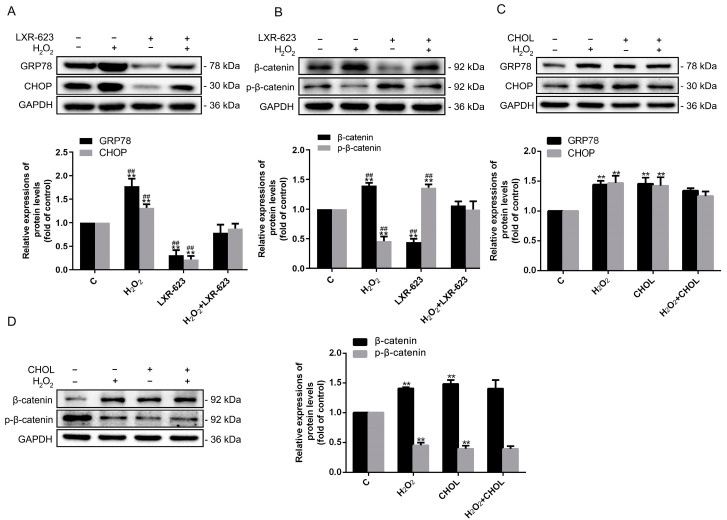
LXR-623 can inhibit ER stress and the Wnt/β-catenin pathway by promoting cholesterol efflux, and adding cholesterol extracellularly can induce ER stress and activate the Wnt/β-catenin pathway by increasing the deposition of intracellular cholesterol. (**A**–**D**) The expressions of GRP78, CHOP, β-catenin, and p-β-catenin were detected by Western blot. *n* = 3, ** *p* < 0.001 vs. the C group; ^##^ *p* < 0.001 vs. the H_2_O_2_ + LXR-623 group. The C (control) group of LXR-623 and cholesterol were treated by DMSO and chloroform, respectively; H_2_O_2_, cells treated with H_2_O_2_ (500 μmol/L) for 24 h; LXR-623, cells treated with LXR-623 (5 μmol/L) for 24 h; H_2_O_2_ + LXR-623, cells treated with LXR-623 (5 μmol/L) for 2 h and then treated with H_2_O_2_ (500 µmol/L) for 22 h; CHOL, cells were treated by cholesterol (100 μmol/L) for 24 h; H_2_O_2_ + CHOL, cells were firstly treated by cholesterol (100 μmol/L) for 2 h, and then H_2_O_2_ (500 µmol/L) was used to treat cells for 22 h.

**Figure 6 ijms-24-05939-f006:**
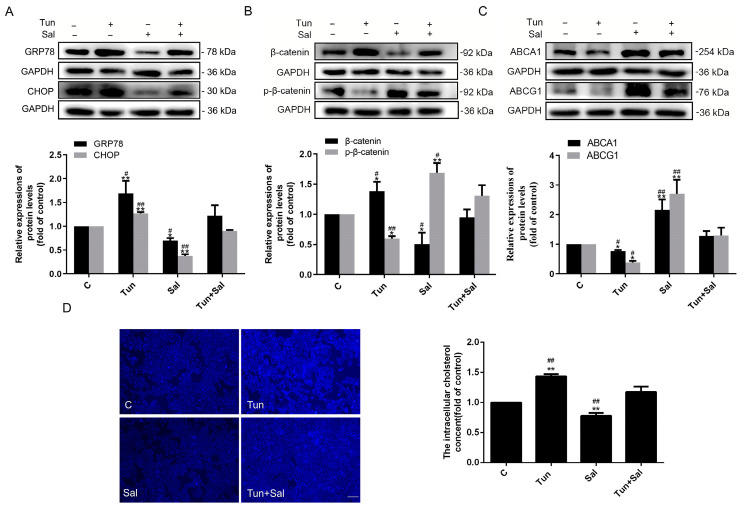
Activating ER stress can increase the accumulation of cholesterol and activate Wnt/β-catenin pathway, and inhibition of the Wnt/β-catenin pathway can attenuate the accumulation of cholesterol and ER stress. (**A**–**C**) The expressions of GRP78, CHOP, β-catenin, p-β-catenin ABCA1, and ABCG1 were detected by Western blot. (**D**) Filipin staining detected the intracellular cholesterol of HUVECs (magnification 5×). *n* = 3, * *p* < 0.05 and ** *p* < 0.001 vs. the C group; ^#^ *p* < 0.05 and ^##^ *p* < 0.001 vs. the Tun + Sal group. C, control group (treated by DMSO); Tun, cells treated with tunicamycin (10 μmol/L) for 24 h; Sal, cells treated with salinomycin (5 μmol/L) for 24 h; Tun + Sal, cells treated with tunicamycin (10 μmol/L) for 2 h, and then salinomycin (5 μmol/L) were used to treat cells for 22 h. The scale of the images is 100 μm.

**Figure 7 ijms-24-05939-f007:**
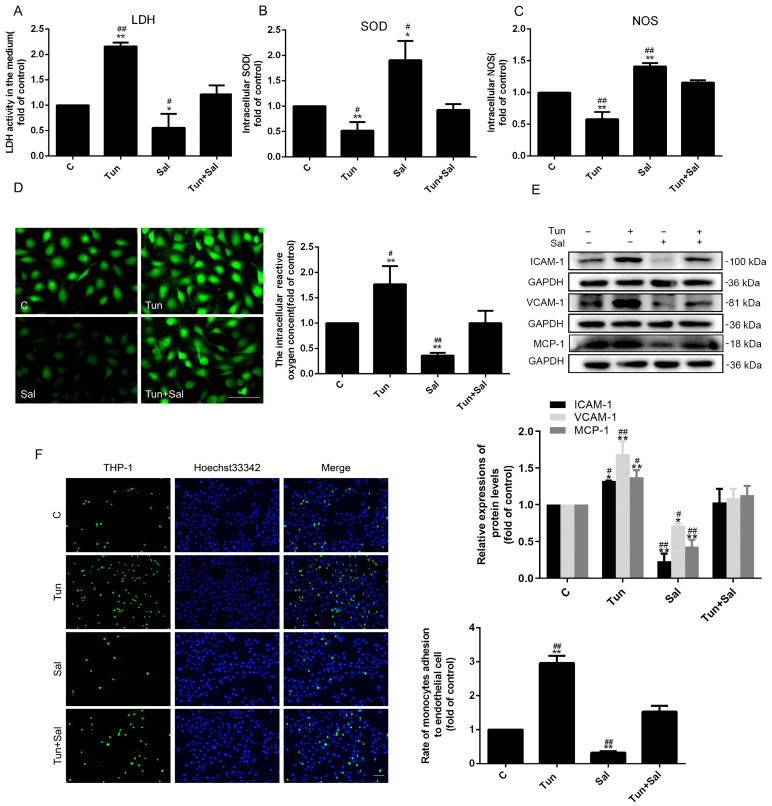
Activating ER stress can increase HUVEC cell damage and adhesion capacity, and inhibiting Wnt/β-catenin pathway can attenuate HUVEC cell damage and adhesion capacity. (**A**) LDH, (**B**) SOD, and (**C**) NOS activity were detected by an enzyme activity kit. (**D**) The activity of ROS was detected by a DCFH-DA fluorescemce probe (15 μmol/L), 10× magnification. (**E**) The expressions of ICAM-1, VCAM-1, and MCP-1 were detected by Western blot. (**F**) The adhesion capacity of HUVEC was assessed by the number of attaching THP-1 (magnification 4×). *n* = 3, * *p* < 0.05 and ** *p* < 0.001 vs. the C group; ^#^ *p* < 0.05 and ^##^ *p* < 0.001 vs. the Tun + Sal group. C, control group (treated by DMSO); Tun, cells treated with tunicamycin (10 μmol/L) for 24 h; Sal, cells treated with salinomycin (5 μmol/L) for 24 h; Tun + Sal, cells treated with tunicamycin (10 μmol/L) for 2 h and then treated with salinomycin (5 μmol/L) for 22 h. The scale of the images is 100 μm.

**Figure 8 ijms-24-05939-f008:**
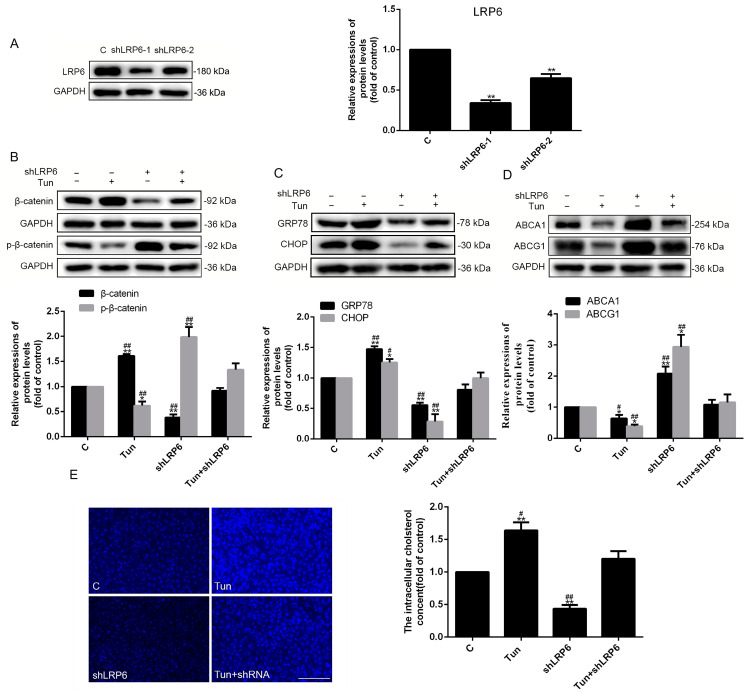
Suppressing the Wnt/β-catenin signaling pathway by knocking down LRP6 specifically has the same functions as salinomycin, that is, inhibiting ER stress and lowering cholesterol accumulation. (**A**–**D**) The expressions of GRP78, CHOP, β-catenin, and p-β-catenin ABCA1 and ABCG1 were detected by Western blot. (**E**) Filipin staining detected the intracellular cholesterol of HUVECs, 10× magnification. *n* = 3, * *p* < 0.05 and ** *p* < 0.001 vs. the C group; ^#^ *p* < 0.05 and ^##^ *p* < 0.001 vs. the Tun + shLRP6 group. C, control group (treated by DMSO); Tun, cells treated with tunicamycin (10 μmol/L) for 24 h; shLRP6, cells were infected with lentivirus containing LRP6-specific knockout for 48 h; Tun + shLRP6, cells were continuously treated with tunicamycin (10 μmol/L) for 24 h after transfection with lentivirus containing LRP6-specific knockout for 48 h. The scale of the images is 100 μm.

**Figure 9 ijms-24-05939-f009:**
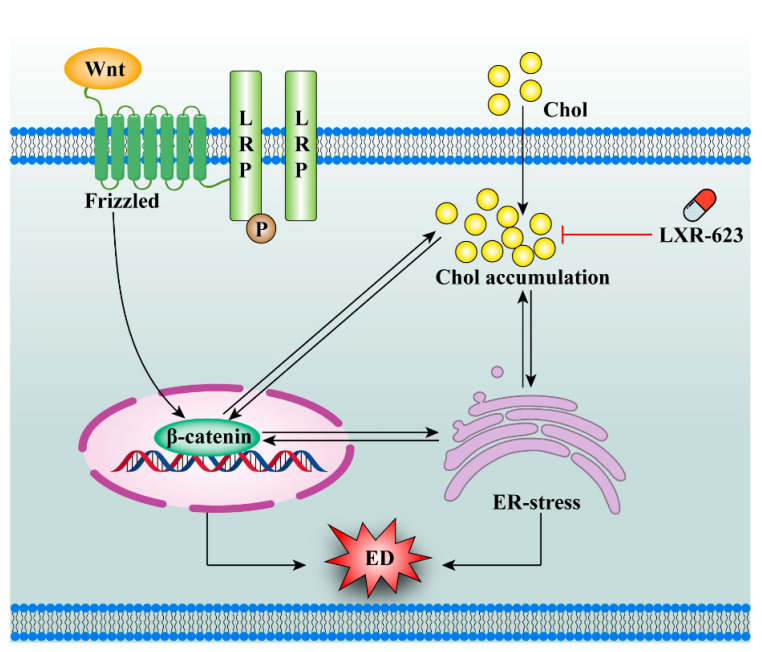
The possible relationship between ER stress, cholesterol metabolism, and the Wnt/β-catenin pathway.

## Data Availability

Data will be made available upon request.

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
