# Peer review of "The Effect of Cholesterol Efflux on Endothelial Dysfunction Caused by Oxidative Stress"

_ijms, 2023, doi:10.3390/ijms24065939_

Round 1

Reviewer 1 Report

„The effect of cholesterol efflux on endothelial dysfunction caused by oxidative stress“

In the manuscript „The effect of cholesterol efflux on endothelial dysfunction caused by oxidative stress“ the authors aimed to investigate the effect of cholesterol efflux on endothelial dysfunction caused by oxidative stress and its correlation to endoplasmatic reticulum stress and Wnt/ß-catenin pathway using H2O2 cell culture system.

This reviewer has major concerns about the study outline and the quality of the provided results as listed in the following sections.

Major points:

-        Research articles using only one type of cell line for the experiments are not convincing, since it is not clear whether these are general reported effects or just an artefact of the used single cell line. Therefore, the reported key experiments need to be repeated on 1-2 more cell lines or primary isolated cells.

-        Study presentation: In general, the presentation, description and the used statistical testing is not convincing. From the description of the experiments, it is not clear whether n=3 is the number of samples or the number of independent experiments (here the number of replicates per experiment is missing.)

-        Methods and figure legends, Statistical analysis: Information concerning the normality testing are missing. Continuous data should be expressed as mean ± SD or median and range, according to normality of distribution. A description of the data presentation is missing: mean or median (which is depending on the normality testing)? SD or SEM?

-        Methods and figure legends, Statistical analysis: This reviewer is not convinced of the selected statistical analysis. There is no information on the t-test used. Furthermore, the reviewer is of the opinion that a one-way ANOVA test should be carried out since the different groups are compared with each other.

-        Methods: From the description of the experiment set up it is not clear if or how controls were treated. Did authors use placebo, vehicle, solvent etc?

-        In a recently publication from Ransy et al. [1] in IJMS the authors described the problem of using H2O2 cell culture models with disproportionally high concentrations of H2O2 with regard to physiological oxygen concentrations. Here in the submitted manuscript, the authors used such a concentration of 500  mmol/L. The authors should consider the reported results in their own study and manuscript.   

-        The passage number of the used HUVECs is missing, which is an important aspect since the passage number should not exceed more than ten for optimal results [2]

-        Please add description of analysis of the fluorescence staining to relevant paragraphs. From the provided information, it is not clear how the authors performed the quantification of the fluorescence data. How many image sections were analyzed?

-        Add confluency/density of the used cells in paragraph 4.3. Assessment of the ED and to paragraph 4.4. Measurement of ROS contents. The confluency of HUVECS when adding DCFH-DA working solution is not described.

-        For assessment of the ED, it is not outlined what kind of assays were performed. Did authors follow the kits protocol?

-        Results Western Blot: Protein band sizes need to be included. In addition, it is not described how the protein expression is determined. Are the protein bands normalized to GAPDH?

-        Please provide original uncropped images of western blot membranes as supplementary material as suggested by MDPI in Instructions for authors: “In order to ensure the integrity and scientific validity of blots (including, but not limited to, Western blots) and the reporting of gel data, original, uncropped and unadjusted images should be uploaded as Supporting Information files at the time of initial submission” The authors provided cropped gel bands without protein sizes, which is not acceptable.

-        Please add scale bars to pictures of cells and replace pictures of filipin staining with more sharp pictures.

-        Results: Authors should always perform a DAPI staining together with other fluorescence dyes. It is unclear, how many HUVECS there are in an image section

-        As authors mean to explore cell damage, they should perform vitality/viability tests, and analyze the p53 signaling and the DNA Damage response pathway. Authors should also perform analysis on mitochondrial and DNA damage and on secreted factors.

-        English needs to be improved by a native speaker

Minor points

-        Figure 4 D column chart inscription of y-axis is wrong

-        Explain all abbreviations (e.g. ABCA1, LXRa, ABCG1) and use same abbreviation when already introduced (e.g. Fig 6: T/Tun and S/Sal

-        Please control space character in the text. There are several spaces missing and in line 51 is a double space

-        Please sort pictures and corresponding graphs in figures the same way. Sometimes graphs are aside, sometimes beneath the images (Fig. 3 + 7)

[1] Ransy, C., Vaz, C., Lombès, A., & Bouillaud, F. (2020). Use of H2O2 to Cause Oxidative Stress, the Catalase Issue. International journal of molecular sciences, 21(23), 9149. https://doi.org/10.3390/ijms21239149

[2] Liao, H., He, H., Chen, Y., Zeng, F., Huang, J., Wu, L., & Chen, Y. (2014). Effects of long-term serial cell passaging on cell spreading, migration, and cell-surface ultrastructures of cultured vascular endothelial cells. Cytotechnology, 66(2), 229–238. https://doi.org/10.1007/s10616-013-9560-8

Reviewer 2 Report

This Manuscript by Ye and colleagues investigates the interaction and contribution of ER stress, cholesterol accumulation and involvement of Wnt/B-catenin pathway in development of endothelial dysfunction in cultured endothelial cells. The studies are interesting and exploration of contribution of these factors, and their interconnection at the molecular level provides insights towards understanding the molecular basis of ED.

Data provided are mostly solid and support conclusion. Nevertheless, there are issues that need addressing as follows:

1- Major conclusions mostly rely on the use of pharmacological agonist/antagonists, which although valid, may have some non-specific effects, and thus if could be supported by knock down or overexpression, would greatly strengthen the conclusions.  For example the H2O2-induced decreased levels of LXRa are observed concomitantly with decreased levels of ABCA1 and ABCG1.  An agonist of LXRa (LXR-623) overcomes this effect. Based on these results, it is concluded that the ABCA1 and ABCG1 reduction is due to decreased levels of LXRa, and consequently its decreased activity. Although the observation is supportive of this hypothesis, is not conclusive. The observed decrease may be association but not cause and effect and LXR-623 may have non-specific effect. However, such conclusion and those from other experiments that uses this agonist will be strengthened if LXRa is exogenously overexpressed in cells and shown to overcome the effects of H2O2.  

Another example is the use of pharmacological agonist/antagonist to induce ER stress and inhibit Wnt/B-catenin pathway and thus explore Wnt/B-catenin pathway contribution to ED by determining the effect on cholesterol efflux proteins (ABCA1 and ABCG1)  and ED markers (LDH, SOD, NOS). However, a knock down strategy for B-catenin in these experiments  would greatly strengthen the conclusion and will provide additional insight regarding whether it is the level, or modification (reduced phosphorylation), of B-catenin is responsible for the observed effect. Specifically since these two show opposing response to H2O2 or other inducer of ER stress.

2- Figure 2C - the two efflux related proteins had opposing response to cholesterol? Any reason/ hypothesis for this observation?

Also the conclusion for experiments described in Fig. 2C and D ( (lines 109-111). is not entirely consistent with the presented results. Combined treatment appears to significantly reduce the increase in total cholesterol that is observed with H2O2 alone (Fig. 2D). Also the significant increase in ABCG1 that is observed with cholesterol alone appears to be significantly and drastically reduced in combined treatment (Fig. 2C).

3- Figure 3- Levels of changes in LDH and SOD activities in response to H2O2 between the two sets of experiments presented in Figures 3A and B, compared to 3E and F, are vastly different. In Figure 3A LDH activity is shown as increased by 2 fold, while similar H202 treatment in figure 3E appears to have increased it by ~ 8 fold. The SOD reduction in figure 3B appears to be ~25% while in 3F it appears to be ~80%. Is there any explanation for this vast variation in response? such huge variations will make it difficult to conclude that other changes are outside experimental variations and statistically meaningful.

Also in this set of experiments treatment with combination of cholesterol and H202 appears to ameliorate the effect of H020. There is a statistically significant difference between the combined treatment and H202 treatment alone. This is not consistent with the stated conclusion (lines 135-137).

4- Figure 4 – From the results the authors conclude that there was no difference in the levels of adhesion capacity when comparing the effect of H202 or cholesterol alone compared to combination, and “Hence, the adhesion ability of HUVECs could be improved through treated the 164 cholesterol extracellularly” (lines 161-165). How do the authors explain this observation? Hypothesis? Could the conclusion be that the increased adhesion specifically in response to H2O2 would be ameliorated by exogenous cholesterol treatment?

5- Figure 5 - The increase in GRP78 and CHOP in response to H2O2 treatment appears very modest, and especially the GAPDH is highly overexposed thus making it difficult to have a reasonably accurate normalization. This also applies to Figure 5B.  Also For experiments described in this section A more relevant conclusion than what is stated in lines 193-194 that incorporate the results of H2O2, cholesterol and combination treatments would be helpful.

6- Methods- Some additional information is needed in this section.

-  In Cell culture section, regarding HUVECS, what passage and what other factors beside ECM and 10% FBS were used. There was no endothelial growth factor?

- In Filipin staining section what is meant by co-cultured? is this referring to co-culture with THP-1 cells?

- In Measurement of ROS content section how were the quantifications carried out?

7- The manuscript requires major editorial modifications.

-Also discussion section needs to be significantly streamlined and focused on the results and specifically address areas of unexpected finding. For instance potential hypothesis regarding why H202 and cholesterol combination did not acquire synergistic effect. As it stands two pages of continuous block of writing with only two paragraphs for the entire discussion, is hard to follow. It does not bring out a focused and clear discussion of the results.

Round 2

Reviewer 1 Report

Response Reviewer 1:

Not all questions were answered satisfactorily. There is still no information about testing the normal distribution in the Statistics section. 

Major points:

- Research articles using only one type of cell line for the experiments are not convincing, since it is not clear whether these are general reported effects or just an artefact of the used single cell line. Therefore, the reported key experiments need to be repeated on 1-2 more cell lines or primary isolated cells.

Response to the reviewer: Yes, it would be better if use the other cell lines. However, the HUVEC is a primary cell and there are no other same cell lines. In fact, there are many researches basically only used HUVEC in their papers. For example, the reference as follow:

1.Li W, Wang C, Zhang D, Zeng K, Xiao S, Chen F, Luo J. Azilsartan ameliorates ox-LDL-induced endothelial dysfunction via promoting the expression of KLF2. Aging (Albany NY). 2021 ;13(9):12996-13005.

2.Niu C, Chen Z, Kim KT, Sun J, Xue M, Chen G, Li S, Shen Y, Zhu Z, Wang X, Liang J, Jiang C, Cong W, Jin L, Li X. Metformin alleviates hyperglycemia-induced endothelial impairment by downregulating autophagy via the Hedgehog pathway. Autophagy 2019;15(5):843-870.

Response to the authors:

Fine. Unfortunately, the authors should include a limitation statement at the end of the conclusion that describes that the results need to be interpreted with caution, since the provided results are raised from only one primary cell line.  

Alternative for HUVEC: An alternative that is also becoming popular Human Microvasculature Endothelial Cells (HMEC). These are often considered more similar to the actual in vivo models since they are not derived from the very specific umbilical cord tissue.

- Study presentation: In general, the presentation, description and the used statistical testing is not convincing. From the description of the experiments, it is not clear whether n=3 is the number of samples or the number of independent experiments (here the number of replicates per experiment is missing.)

Response to the reviewer: In Statistical analysis of the methods we have described that all treatment group have been repeated three times, possible, it is not enough clearly, and we have revised it in the manuscript.

Response to the authors:

The authors wrote in their statistical analysis section, ”Each treatment group of HUVEC had at least three replicates (n=3)”. This means that each experimental group was performed with three replicates. However, there is no specification how many independent experiments the authors performed. Please clearly describe how many independent experiments with the number of replicates were performed.

- Methods and figure legends, Statistical analysis: Information concerning the normality testing are missing. Continuous data should be expressed as mean ± SD or median and range, according to normality of distribution. A description of the data presentation is missing: mean or median (which is depending on the normality testing)? SD or SEM?

Response to the reviewer: We have revised them in the manuscript.

 Response to the authors:

The information concerning a testing of normality is still missing in the statistical analysis section. Please see below.

A normality test is necessary to determine whether sample data has been drawn from a normally distributed population (within some tolerance). A number of statistical tests, such as the Student's t-test and the one-way and two-way ANOVA, require a normally distributed sample population. Therefore the authors need to test their data for normality distribution.

- In a recently publication from Ransy et al. [1] in IJMS the authors described the problem of using H2O2 cell culture models with disproportionally high concentrations of H2O2 with regard to physiological oxygen concentrations. Here in the submitted manuscript, the authors used such a concentration of 500 mmol/L. The authors should consider the reported results in their own study and manuscript.

Response to the reviewer: Firstly, I am very sorry for making a mistake. In our studies, the culture medium were the DMEM and not the ECM medium. Therefore, we corrected the mistake in the manuscript. Additionally, the concentration of H2O2 were 400-500mmol/L in most published papers about HUVEC. For example, the reference as follow. Interestingly, in the study of Xing et al’s, the concentration of H2O2 reached 2000 mmol/L.

1. _Liang X, Lin F, Ding Y, Zhang Y, Li M, Zhou X, Meng Q, Ma X, Wei L, Fan H, Liu Z. Conditioned medium from induced pluripotent stem cell-derived mesenchymal stem cells accelerates cutaneous wound healing through enhanced angiogenesis. Stem Cell Res Ther. 2021 May 20;12(1):295. doi: 10.1186/s13287-021-02366-x. PMID: 34016178; PMCID: PMC8139053.

2. Xing X, Li Z, Yang X, Li M, Liu C, Pang Y, Zhang L, Li X, Liu G, Xiao Y. Adipose-derived mesenchymal stem cells-derived exosome-mediated microRNA-342-5p protects endothelial cells against atherosclerosis. Aging (Albany NY). 2020 ;12(4):3880-3898.

3. Liao L, Gong L, Zhou M, Xue X, Li Y, Peng C. Leonurine Ameliorates Oxidative Stress and Insufficient Angiogenesis by Regulating the PI3K/Akt-eNOS Signaling Pathway

in H2O2-Induced HUVECs. Oxid Med Cell Longev. 2021 Aug 3;2021:9919466. doi: 10.1155/2021/9919466. PMID: 34394836; PMCID: PMC8357476.

Response to the authors:

This reviewer does not doubt that numerous other papers also use disproportionally high concentrations of H2O2 as mentioned in the publication from Ransy et al. [1] in IJMS. This was also noted in the publication of Ransy et al.. The authors cannot ignore this publication, which appeared in the same journal IJMS. Therefore, they should critically discuss this aspect in their discussion or show additional experiments that the results are comparable with low physiological H2O2 concentrations.

- The passage number of the used HUVECs is missing, which is an important aspect since the passage number should not exceed more than ten for optimal results [2]

Response to the reviewer: The HUVECs were cultured with DMEM medium. Theoretically, this cell can be cultured for many passages, but we do experiments within 30 passages.

Response to the authors:

Passage number is still not integrated in the HUVEC cell culture section.

- Please add description of analysis of the fluorescence staining to relevant paragraphs. From the provided information, it is not clear how the authors performed the quantification of the fluorescence data. How many image sections were analyzed?

Response to the reviewer: We have described the fluorescence staining in the section of Filipin staining, Measurement of ROS contents and Adhesion Ability of HUVECs to THP-1 Cells. Moreover, the quantification of the fluorescence data was clearly described in relevant paragraphs.

Response to the authors:

Yes, the authors have described how they have performed the staining. Nevertheless, a clear description how they have performed the quantification is missing (no description in the methods section or figure legends).

-   Where background mean intensities determined and substrated from the mean values?

-   Was a normalization performed?

-   How many cells or areas were measured?

Round 3

Reviewer 1 Report

Thanks for revising the manuscript according to the addressed points. Please edit the last comment.

Major points:

-       The information of the used test method (for example: D’Agostino Pearson, Shapiro Wilk or Kolmogorov-Smirnov) of normality is still missing in the statistical analysis section. Please see below.

L490/491:

All data, from normally distributed populations, were expressed as mean±standard deviation.

Author Response

Response to the reviewer: We have revised them in the manuscript.